# A whole lifespan mouse multi-tissue DNA methylation clock

**Margarita V Meer[1], Dmitriy I Podolskiy[1], Alexander Tyshkovskiy[1,2], Vadim N Gladyshev[1]***

[1]Division of Genetics, Department of Medicine, Brigham and Women's Hospital and Harvard Medical School, Boston, United States; [2]Center for Data-Intensive Biomedicine and Biotechnology, Skolkovo Institute of Science and Technology, Moscow, Russia

**Abstract** Age predictors based on DNA methylation levels at a small set of CpG sites, DNAm clocks, have been developed for humans and extended to several other species. Three currently available versions of mouse DNAm clocks were either created for individual tissues or tuned toward young ages. Here, we constructed a robust multi-tissue age predictor based on 435 CpG sites, which covers the entire mouse lifespan and remains unbiased with respect to any particular age group. It can successfully detect the effects of certain lifespan-modulating interventions on DNAm age as well as the rejuvenation effect related to the transition from fibroblasts to iPSCs. We have carried out comparative analyses of available mouse DNAm clocks, which revealed their broad applicability, but also certain limitations to the use of tissue-specific and multi-tissue age predictors. Together, these tools should help address diverse questions in aging research.
DOI: https://doi.org/10.7554/eLife.40675.001

## Introduction

A robust and precise marker, which can be used to estimate the biological age of organisms and to evaluate the effect of different interventions on lifespan has been a Holy Grail of aging research since the first days of this field. Various biomarkers of age have previously been suggested, which are based on telomere length (*Harley et al., 1990*), mutation accumulation (*Dollé et al., 2000*; *Podolskiy et al., 2016*; *Podolskiy and Gladyshev, 2016*), gene expression levels (*de Magalhães et al., 2009*) or T-cell-specific DNA rearrangements (*Zubakov et al., 2010*). However, these approaches to construct a precise biomarker of age proved to be relatively limited in their potential to assess the aging process and examine its modulation by various interventions, mostly due to large variability of detected ages. In contrast, analyses of DNA methylation provided an opportunity for much more accurate estimation of age of organisms. This method is based on the chronological age of subjects and reflects positive or negative changes which are associated with lifespan extension or shortening. Such biomarkers do not measure the biological age in a direct way but may reveal a change in it.

Age predictors based on DNA methylation were first proposed for human saliva samples (*Bocklandt et al., 2011*) and were later developed into robust blood-specific (*Hannum et al., 2013*) and multi-tissue (*Horvath, 2013*) biomarkers of age. To date, age predictors based on systematic changes of DNA methylation with age, DNAm clocks, remain the most precise markers of age. For example, the human multi-tissue DNAm clock has a median error of 3.6 years (*Horvath, 2013*), which does not exceed ~3% of maximal human lifespan. Additional versions of human DNAm clock are being developed that can address broad medical and public health questions related to aging. The already developed human DNAm clocks reflect age acceleration associated with genetic background, such as progeria (*Maierhofer et al., 2017*), as well as caused by lifestyle and diet, such as

*For correspondence:
vgladyshev@rics.bwh.harvard.edu

Competing interests: The authors declare that no competing interests exist.

smoking and obesity (*Quach et al., 2017*; *Levine et al., 2018*). Age predictors based on DNA methylation were also recently constructed for whales (*Polanowski et al., 2014*), wolves and dogs (*Thompson et al., 2017*). Availability of these clocks may help to speed up testing lifespan-extending interventions, such as rapamycin (*Urfer et al., 2017*). It remains unclear, however, whether the best clocks should utilize a few key CpG sites, a large group of sites or a change of the pattern of global methylation.

Application of DNAm clocks to assess the DNA methylation age would be especially useful when applied to common model organisms of aging. In this regard, three DNAm clock versions have been developed for mice, including blood-based (*Petkovich et al., 2017*) and liver-based (*Wang et al., 2017*) clocks, and a young age-specific multi-tissue clock (*Stubbs et al., 2017*) (*Table 1*). These clocks have been already used to test the effects of several genetic, dietary or pharmacological interventions (*Petkovich et al., 2017*; *Wang et al., 2017*; *Stubbs et al., 2017*; *Hahn et al., 2017*; *Cole et al., 2017*) on mouse lifespan. However, the existing mouse DNAm clocks have a limitation of tissue specificity, and it also remains unclear how they relate to and perform with respect to each other. Here, we created a DNAm clock based on DNA methylomes of different tissues collected from mice aged from 1 week to 35 months old. We further performed comparative analysis of previously published DNAm age estimators and the newly constructed DNAm multi-tissue clock.

## Results

### Construction of a multi-tissue clock

A mouse DNAm multi-tissue clock (YOung Multi-Tissue or YOMT in what follows) was recently developed (*Stubbs et al., 2017*), which was tuned for tissues of younger mice, for example its mean absolute error, MAE, increased from 2.14 to 4.66 weeks if applied to animals of <20 weeks and 20–40 weeks of age, respectively. In addition, application of this clock to new samples required normalization of the data to the original dataset used to construct YOMT (*Stubbs et al., 2017*). We aimed to create a multi-tissue age predictor, which can be easily employed by the community to estimate the DNA methylation age of samples of all accessible chronological ages (Whole Lifespan Multi-tissue or WLMT clock in what follows).

To develop the clock, we first collected publicly available data on DNA methylation in mice (*Supplementary file 1*). The combined dataset was strongly biased toward younger chronological ages, with the blood being the only tissue well represented by different ages. Therefore, to construct the methylation clock based on comprehensive compendium of samples, we additionally performed reduced representation bisulfite sequencing (RRBS) of 76 liver, lung, brain and heart samples of 6-, 10-, 12-, 20- and 30-month-old C57Bl/6 male mice, with 4–5 biological replicates per tissue per time point, except for two age/tissue points (*Supplementary file 2*).

While in theory whole genome bisulfite sequencing (WGBS) should include all CpG sites obtained by any other enrichment method, in reality the overlap between the results of WGBS and RRBS is such that only ~20% of CpG sites obtained by RRBS are sufficiently covered by WGBS if the same samples are sequenced by both methods (*Harris et al., 2010*). Consequently, pooling together methylation profiles obtained by the two methods resulted in too few sites covered in all pooled samples. Also, even though the use of WGBS by the research community to characterize DNA

**Table 1.** Summary of available mouse DNAm age predictors.

| Tissue type | Reported precision, $R^2$ | Age range of training set, months | Number of CpG sites in the model | Publication |
|---|---|---|---|---|
| Blood | > 0.9008 | 3.–35 | 90 | (*Petkovich et al., 2017*) |
| Liver | 0.91 | 0.2–26 | 148* | (*Wang et al., 2017*) |
| Multi-tissue | 0.7 | 0.2–9.5 | 329 | (*Stubbs et al., 2017*) |
| Multi-tissue | 0.89 | 0.2–35 | 435 | This paper |

*Unlike other DNAm clocks, the liver clock is based on a combination of methylation levels on both strands and thus it corresponds to 296 genomic positions.
DOI: https://doi.org/10.7554/eLife.40675.002

methylation becomes more common, the number of available datasets obtained for whole genomes remains relatively limited. For this reason, we used only RRBS data to construct the multi-tissue DNAm clock.

We filtered out RRBS samples with fewer than $2 \cdot 10^6$ CpG sites covered (*Figure 1-figure supplement 1*) as well as those with the signatures of batch effects (*Figure 1-figure supplement 2*). To build a multi-tissue DNAm clock, we used untreated (i.e. no interventions, mutations or other conditions) wild-type (C57Bl/6) samples. The resulting combined dataset of DNA methylomes included 416 samples representing 11 different tissues and cell types with 193,439 sites covered in all of them. The dataset was then divided into training (80%) and test (20%) subsets with stratification based on tissue types. We then performed elastic net regression with 10-fold cross-validation on DNA methylomes pooled together. The WLMT linear model was defined as follows: $age(days) = w_1 \cdot_1 + w_2 \cdot_2 + \ldots + w_{435} \cdot_{435} + 234.64$, where $w_i$ is weight for the CpG site $i$ and $_i$ is percent methylation at the site $i$. The resulting DNAm clock included 435 CpG sites (*Figure 1, 2*). The training set showed the MAE of 28.6 days, while the MAE of the test set was 72.7 days. $R^2$ for the training and test sets were 0.98 and 0.89, respectively (*Figure 1*). While WLMT in the younger ages was tested on many tissues, the test set included only two non-blood samples older than 600 days: one lung and one cortex samples both collected for 900-day-old animals, and values of absolute errors for these samples (240 and 381 days) were larger than for the blood samples on similar age (113 days).

The majority of presently available RRBS samples come from male mice. Therefore, WLMT is mostly (84%) based on male samples. We tested how this affects performance on samples of different sexes by comparison of predictions for male and female samples of the same age range (from 56 to 168 days old). MAE was lower for female samples (33 days), while it was 43.7 days for the male samples, but this difference was not significant ($p = 0.37$, two-tailed Mann–Whitney U test). To test whether there is any difference in age acceleration between these two groups, we performed linear regressions between methylation and chronological ages for both (*Figure 1-figure supplement 3*). There was no significant difference between the slopes of the regressions ($p = 0.72$) which can be explained by either the lack of such acceleration or WLMT clock not being able to detect it. At the same time, the difference in intercepts between the two regressions was also not significant ($p = 0.54$). Thus, WLMT does not distinguish between male and female samples and can be applied to both.

## Comparison of existing mouse DNAm clocks

Three different mouse DNAm clocks have previously been reported (*Table 1*), but their performance has not yet been comparatively analyzed. The reported MAE for YOMT was 23.3 days. MAE has not been estimated, but the clocks based on the half of the samples were estimated to have MAE of 1–3 months for young samples (<10 months old) and 5–7 months for older ages. MAE for the 22-month-old liver samples measured by the liver clock was 4.2 months.

The four age predictors that are compared here to the WLMT test set (*Figure 3A*) are biased toward a better performance as our test set includes samples from training sets of these clocks. WLMT was based on samples representing all these datasets combined. Since RRBS results typically include batch effects/dependence on experiment location and procedure, this gives an advantage to WLMT because other clocks were tested on the datasets not represented in their original training sets. As almost all samples turned out to be included into a training set of one clock studied here or another, we utilized different subsets of samples for comparative analyses (*Figure 3B-F*). All untreated wild-type C57BL/6 samples were used for the construction of the blood methylation clock. Thus, it was impossible to compare the blood clock to other age estimators by applying it to the same strain without removing an associated performance bias. Instead, we used control samples from three different studies: growth hormone receptor (GHR) knockout (KO) and Snell dwarf mice as well as caloric restriction applied to B6D2F1 mice (*Petkovich et al., 2017*). In this setup, the methylation clock built using blood samples outperformed all other predictors ($p < 1.9 \cdot 10^{-4}$ for all pairwise comparisons with blood clock, two-tailed Mann–Whitney U test) with the MAE of 2.14 months for the blood clock, 3.76 for WLMT, 6.73 for YOMT and 10.33 for the liver clock (*Figure 3B*). However, the WLMT clock proved to be more precise when applied to a range of different tissues, which

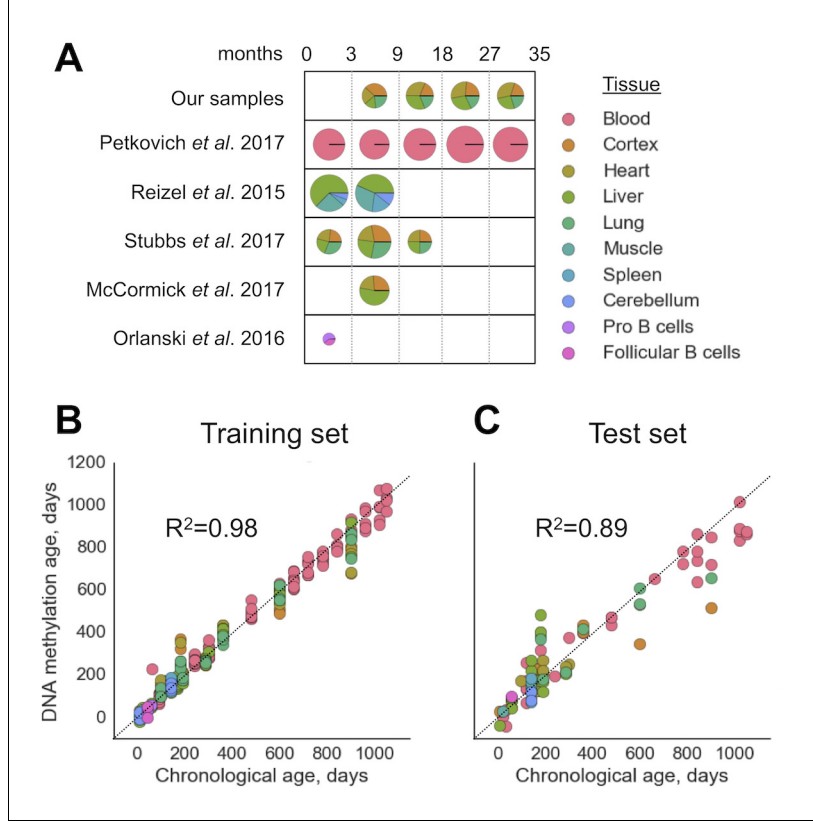

**Figure 1.** Construction of the WLMT clock. (**A**) Datasets used for the clock construction and initial validation. Radii of pie charts correspond to the number of samples in a group. The range of ages is shown above (in months). Performance of the whole-lifespan multi-tissue clock on training (**B**) and test (**C**) sets. MAE on training and test sets was 28.6 and 72.7 days, respectively. Tissues used are indicated on the right according to the color scheme.

DOI: https://doi.org/10.7554/eLife.40675.003

The following figure supplements are available for figure 1:

**Figure supplement 1.** Density plot for the number of CpG sites covered in all samples.

DOI: https://doi.org/10.7554/eLife.40675.004

**Figure supplement 2.** Principle component analysis of samples sequenced in the study.

DOI: https://doi.org/10.7554/eLife.40675.005

**Figure supplement 3.** Performance of the WLMT clock on male (blue) and female (pink) samples.

DOI: https://doi.org/10.7554/eLife.40675.006

**Figure supplement 4.** Number of reads obtained for the sequenced samples.

DOI: https://doi.org/10.7554/eLife.40675.007

excluded blood (*Figure 3E*): in that case, the MAE of WLMT and blood clocks were 2.95 and 5.18 months, respectively ($p = 5.5 \cdot 10^{-5}$, two-tailed Mann–Whitney U test).

On the liver test samples, MAE was 69 days for WLMT and 58 days for YOMT. There were two reasons for a comparatively good performance of YOMT on these samples: first, the dataset included several samples used for training YOMT, and second, YOMT itself was heavily based on liver samples (44% of all samples used for training YOMT were liver samples), unlike WLMT (23% liver samples). Unlike other DNAm-based age predictors discussed here, the liver clock was based on the methylation levels at dinucleotide positions. We used the coordinates of CpG sites contributing to the liver clock (*Wang et al., 2017*) as published without modifications. As in the case of the blood methylation clock, the liver-based predictor performed better than WLMT (*Figure 3-figure supplement 1*) on the samples which were not used in the original training sets of any of the clocks (MAE was 65 days and 244 days, respectively, $p = 0.04$, two-tailed Mann–Whitney U test). However, its performance on these samples was similar to the performance of YOMT (with MAE of 55 days, $p = 0.33$, two-tailed Mann–Whitney U test). It is worth mentioning that there were only three such

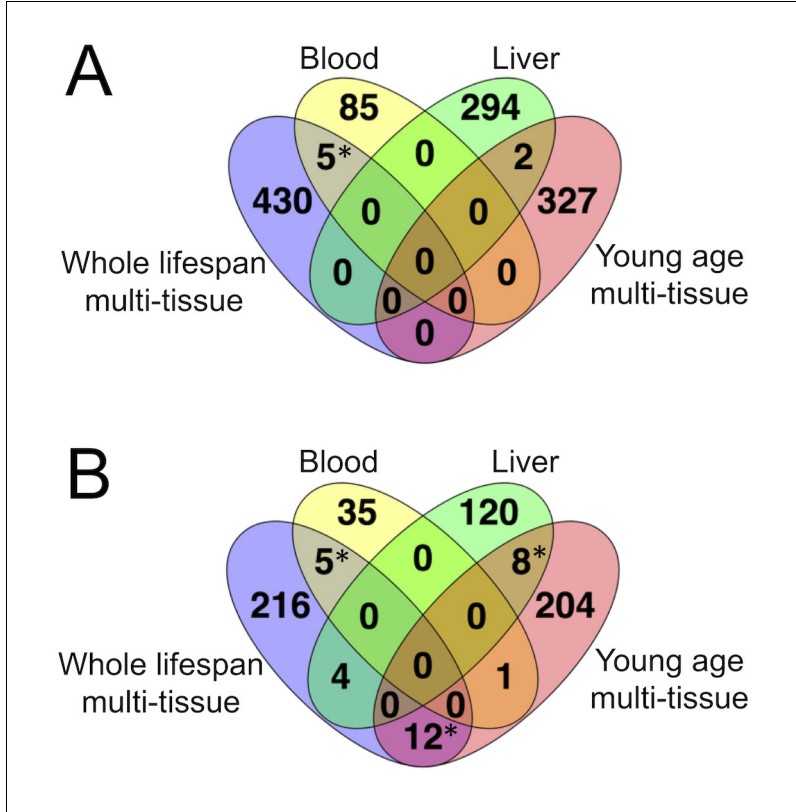

**Figure 2.** Overlap in CpG sites among mouse DNAm clocks. (**A**) overlap in CpG sites. (**B**) overlap in genes containing the clock sites. * - significant with $p<10^{-4}$ (two-tailed Chi-square test with Yates correction).
DOI: https://doi.org/10.7554/eLife.40675.008

The following figure supplements are available for figure 2:

**Figure supplement 1.** Distribution of WLMT clock CpG sites along the mouse genome.
DOI: https://doi.org/10.7554/eLife.40675.009

**Figure supplement 2.** Location of CpG sites forming the four epigenetic clocks in the mouse genome: WLMT sites are shown in blue, YOMT sites in green, blood clock sites in red and liver clock sites in orange.
DOI: https://doi.org/10.7554/eLife.40675.010

**Figure supplement 3.** Frequency of the occurrence of WLMT clock sites in 100 clocks generated based on the robustness study.
DOI: https://doi.org/10.7554/eLife.40675.011

**Figure supplement 4.** Age-related changes in DNA methylation of the ten WLMT clock sites with highest positive weights.
DOI: https://doi.org/10.7554/eLife.40675.012

**Figure supplement 5.** Age-related changes in DNA methylation for the ten WLMT clock sites with lowest negative weights.
DOI: https://doi.org/10.7554/eLife.40675.013

**Figure supplement 6.** Density plots showing distribution of weights of CpG clock sites in the four mouse clocks.
DOI: https://doi.org/10.7554/eLife.40675.014

samples, and performance of WLMT clock on them was significantly lower ($p = 0.02$, two-tailed Mann–Whitney U test) than on other WLMT liver test samples. WLMT was more accurate than the liver-based clocks on non-liver samples (*Figure 3F*) with the MAE of 2.69 months, while the estimated MAE was 11.64 and 14.24 months for YOMT and liver clocks, respectively.

It is important to note that obtaining adequate results for the application of YOMT clock was challenging without the original training set. The obtained DNAm ages ranged from 88.2 to 149.5 days, while the actual range of chronological ages of samples was between 7 and 1050 days, and overall correlation between chronological and DNAm ages was relatively low ($R^2 = 0.36$). This result

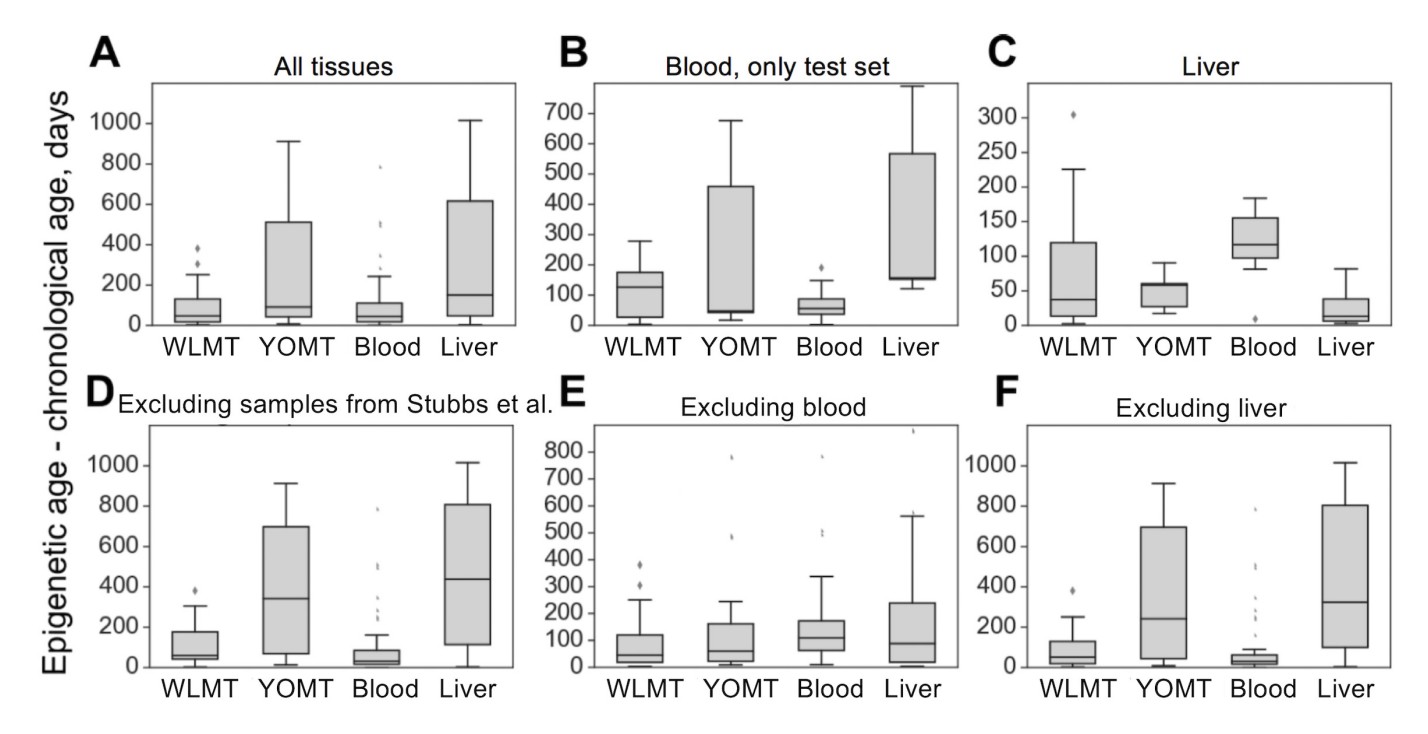

**Figure 3.** Absolute errors of four mouse DNAm clocks applied to untreated wild-type samples. (**A**) In this case, WLMT clock was applied to its test set, whereas for the other age predictors this set also included their training set samples. (**B**) Blood samples which were not used in any of the clocks. (**C**) Only liver samples. (**D**) Samples of all tissues, excluding those used in the YOMT clock. (**E**) Samples of all non-blood tissues. (**F**) All tissues, excluding liver.

DOI: https://doi.org/10.7554/eLife.40675.015

The following figure supplements are available for figure 3:

**Figure supplement 1.** Performance of the WLMT, blood, YOMT and liver clocks on the liver samples not included into the WLMT and liver training sets.

DOI: https://doi.org/10.7554/eLife.40675.016

**Figure supplement 2.** Performance of the WLMT, blood, YOMT and liver clocks.

DOI: https://doi.org/10.7554/eLife.40675.017

was obtained for the samples where more than 90% of clock CpG sites were covered. Thus, it cannot be explained by insufficient site representation (*Supplementary file 1*). This led to a situation where YOMT clock showed good precision on samples with chronological ages ~ 3–5 months old, but lost robustness when applied to samples with more advanced chronological ages (*Figure 3D*). We also noticed and corrected a shift in the coordinates of clock CpG sites defining the blood methylation clock (*Petkovich et al., 2017*) (*Supplementary file 3*), wherein the coordinates were accidently shifted to the next CpG site in a reported supplementary table (*Petkovich et al., 2017*), but not in the analyses. Coordinates of all four clocks used in the current study are reported in *Supplementary file 3*, which can be used for comparative analyses and further applications.

We have also performed a correlation analysis of the estimated methylation ages obtained from the clocks. For this analysis, we only took samples not used in any training set of the studied clocks and having more than 90% of clock CpG sites covered. WLMT correlated best with the blood clock when applied to the blood samples with $r = 0.90$, while on the other tissues it was as low as 0.37. Similarly, the coefficient for WLMT and the liver clock dropped from $r = 0.80$ on liver samples to 0.13 when applied to the other tissues. At the same time, correlation between WLMT and YOMT showed an intermediate value ($r = 0.57$). In addition, we performed a correlation analysis of the delta-age values (i.e. the difference between epigenetic age and chronological age). This analysis showed a pattern which differed from the comparison of absolute values of the DNAm age. Correlation between WLMT and blood clock was 0.42 and 0.63 with the blood and non-blood samples,

respectively. The liver clock correlated with WLMT with $r = 0.81$ on liver samples and $r = 0.63$ on the other samples. Correlation of the delta values of WLMT and YOMT was slightly better than on the absolute values of these clocks, with $r = 0.66$.

In addition, we analyzed 31 heart, lung and hippocampus samples from an independent dataset (*Reizel et al., 2018*) (*Figure 4*). WLMT showed a slightly better performance than the other clocks, with MAE = 53 days while it was 58 days for YOMT, 54 days for the liver clock and 64 days for the blood clock. Interestingly, YOMT and liver clocks produced tight clusters of methylation age points (with standard deviation for YOMT as low as 2 days for young and 3 days for older samples; for the liver clock it was 2 and 8 days, respectively) barely reflecting age change forcing us to conclude that MAE may not be a reliable assessment in this case. To further investigate the comparative behavior of different clocks, we performed linear regression of residual methylation age values against chronological age. Lack of significant slope in this case would be a signature of a clock which does not produce a systematic error and it will be equivalent to having a slope equal to 1 in the linear regression of DNAm age against chronological age. Linear regressions for the residuals in YOMT, blood and liver clocks showed a significant shift of slope from 0. For the WLMT this change was smaller and not significant.

## Application of DNAm age estimators to known lifespan-modulating interventions

We have applied the four mouse DNAm predictors to detect changes in DNAm age associated with the application of longevity interventions (*Figure 5A-F*). As previously reported (*Petkovich et al., 2017*), the blood-based methylation clock reflected the effect of several known lifespan-extending interventions when applied to blood samples (*Figure 5A,B,D,E*), whereas both YOMT and liver-based methylation clocks did not show a significant DNAm age-modulating effect due to the same interventions. WLMT revealed a shift of average DNAm age in all four cases, but with the given sample size, it was statistically significant only for samples with growth hormone receptor knockout (GRH KO, $p = 0.01$, two-tailed Mann–Whitney U test). Both blood ($p = 1.3 \cdot 10^{-6}$, two-tailed Mann–Whitney U test) and WLMT ($p = 8 \cdot 10^{-4}$, two-tailed Mann–Whitney U test) clocks reported the rejuvenation effect associated with the conversion of fibroblasts to induced pluripotent stem cells (iPSCs) (*Figure 5C*). YOMT also showed a shift of methylation age, and even though the effect was significant ($2.1 \cdot 10^{-9}$, two-tailed Mann–Whitney U test), it was small, and the resulting methylation age was still quite high. Liver-based predictors showed the opposite effect with p-value $1.4 \cdot 10^{-8}$, two-tailed Mann–Whitney U test. Only the liver methylation clock reflected a slowdown of aging in Ames dwarf mouse liver samples (*Figure 5F*). However, fewer than 90% of CpG sites used by other DNAm estimators were covered in these samples, while in the other intervention tests all clocks had high sites representation: more than 90% for each sample and more than 95% on average. Another reason for the failure to detect slowdown of aging in these samples might be that DNAm levels were estimated using WGBS. While the liver methylation clock was trained on both WGBS- and RRBS-produced samples, three other models studied here did not use WGBS-produced samples for training and thus they were expected to perform worse on these samples. Overall, our analysis suggests that tissue-specific DNAm clocks for blood and liver samples should be used to assess the DNAm age of these tissues, whereas WLMT clock may be applied also to other tissues. Also, since RRBS-based DNAm clocks do not have good coverage in WGBS samples, RRBS/WGBS-based liver clock is especially recommended for WGBS analyses performed on liver samples and it should be further tested on non-liver WGBS samples.

## WLMT clock sites

The majority of the WLMT clock CpG sites (58%) belong to open sea, 5% to shelves, 11% to shore regions of the genome, and 26% are located within CpG islands. Since we were interested in a common sex-independent signature of aging, the sex chromosomes were excluded from the analysis. Interestingly, all autosomes were represented in the WLMT clock (*Figure 2—figure supplement 1*). Similarly, clock CpG sites were located in all or almost all (in the blood clock) chromosomes (*Figure 2—figure supplement 2*).

We found that none of the WLMT clock sites overlapped with the clock sites of YOMT. The overlaps between blood and WLMT and between liver and YOMT clocks were not statistically significant

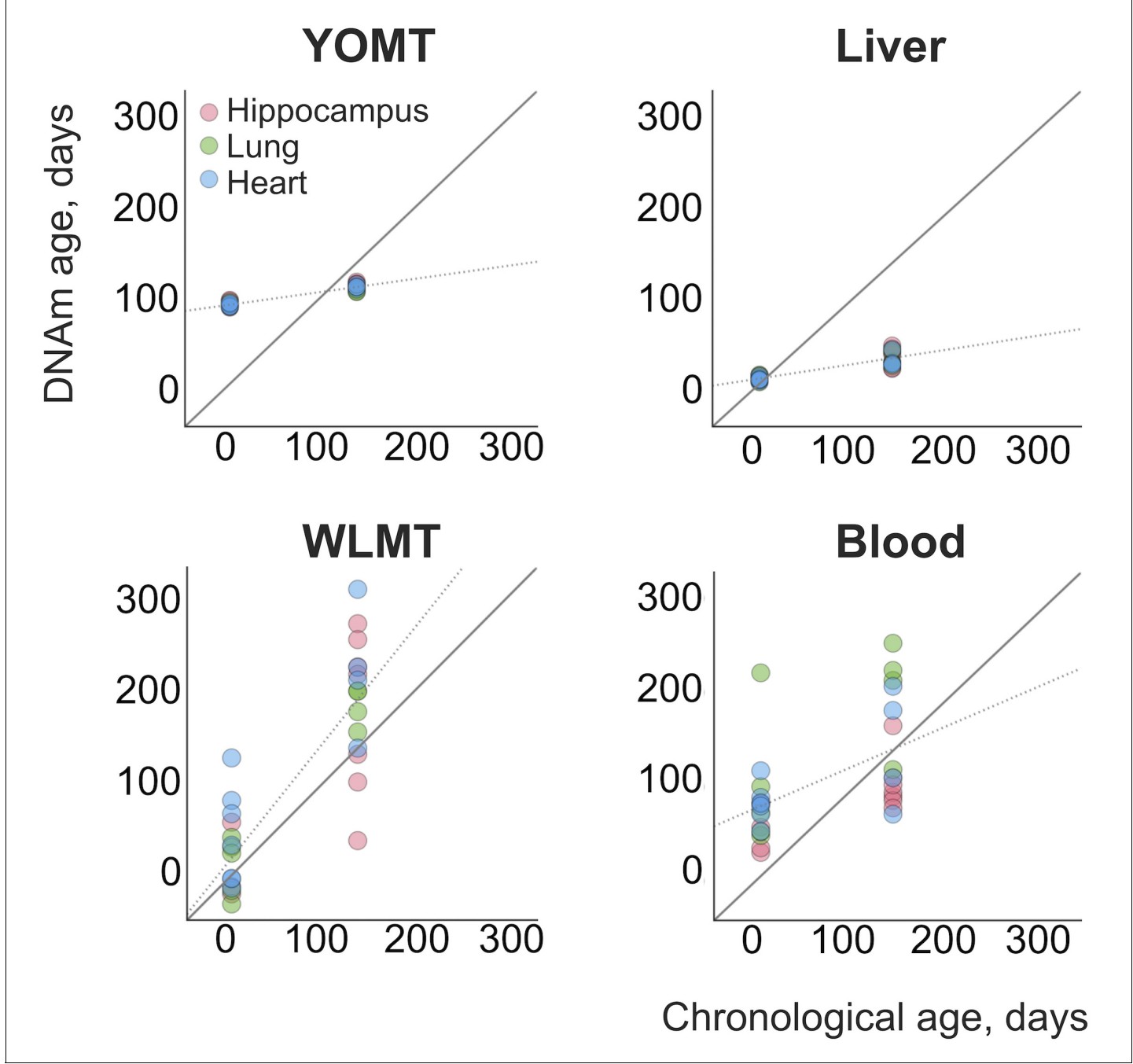

**Figure 4.** DNAm ages estimated for an independent dataset. An additional dataset (*Reizel et al., 2018*) was tested with the four clocks. Solid line represents a perfect correspondence between DNAm age and chronological age. Dashed lines show slopes of linear regressions made for each DNAm clock based on the age values estimated with it. Slopes: 0.14 (YOMT), 0.16 (liver), 1.24 (WLMT), 0.45 (blood). p-values: $p = 1.4 \cdot 10^{-39}$ (YOMT), $p = 2.8 \cdot 10^{-29}$ (liver), $p = 0.12$ (WLMT), $p = 6.4 \cdot 10^{-4}$ (blood).
DOI: https://doi.org/10.7554/eLife.40675.018

(*Figure 2A*). However, there was a significant ($p<10^{-4}$, two-tailed Chi-square test with Yates correction) overlap between genes represented by CpG sites (*Figure 2B*) from two multi-tissue-based predictors, blood and WLMT, liver and YOMT (*Supplementary file 4*). These observations were not surprising because 44% of samples included in the YOMT training set were liver-based, while 38% of the WLMT training set included blood samples.

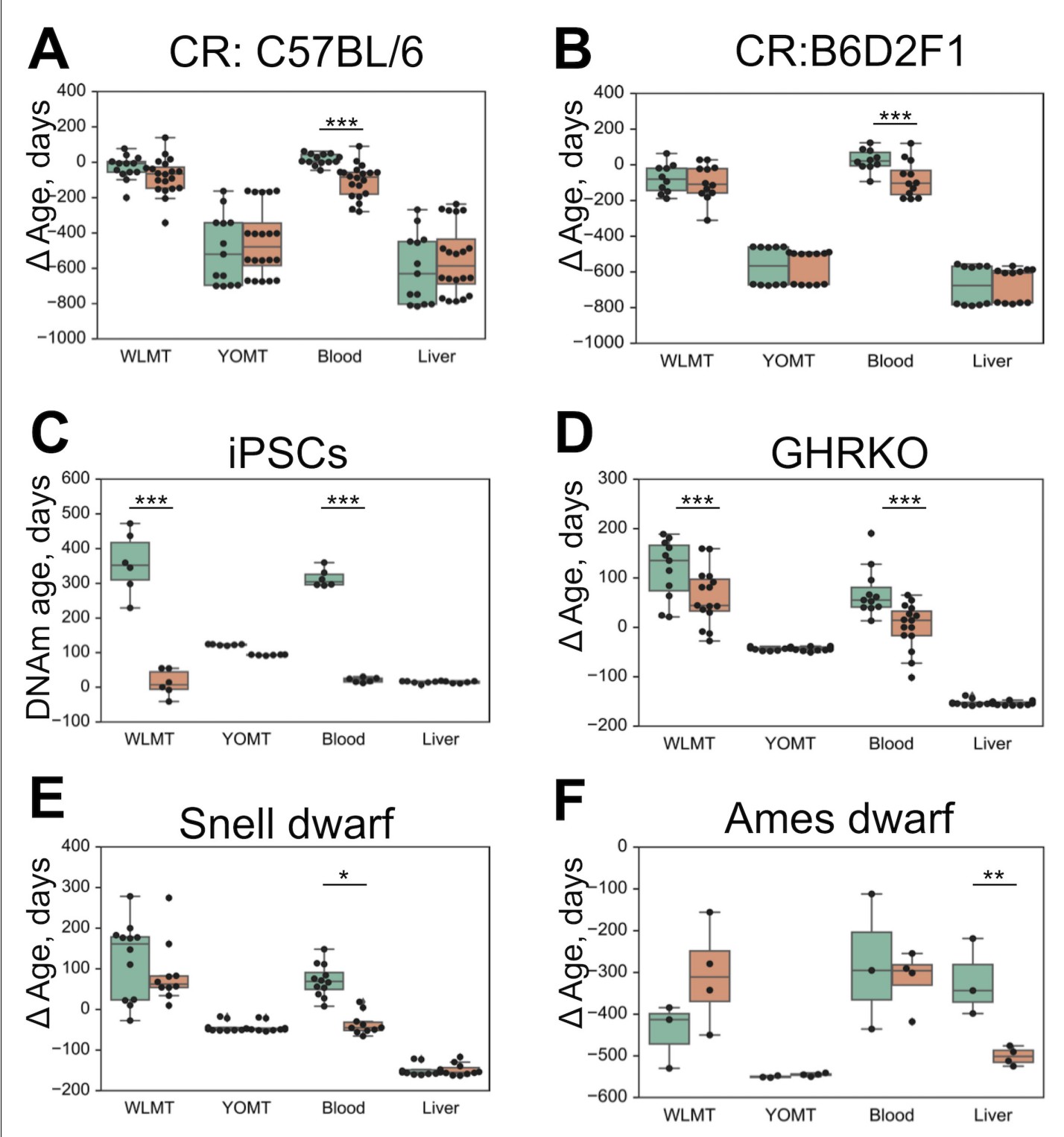

**Figure 5.** Difference in DNAm age estimated by four age predictors and chronological age based on WLMT, blood, YOMT and liver clocks. Untreated and wild-type samples are shown in green, and intervention samples in orange. (**A**) Effect of caloric restriction in mouse strain C57BL/6, DNA from blood. (**B**) Effect of caloric restriction, mouse strain B6D2F1, DNA from blood. (**C**) Conversion of fibroblasts to iPSCs, DNA from cultured cells. (**D**) Growth hormone receptor knockout, DNA from blood. (**E**) Snell dwarf mice, DNA from blood. (**F**) Ames dwarf mice, DNA from liver. ΔAge was defined as the difference between the DNAm age and the chronological age. *- $p < 5 \cdot 10^{-2}$, **- $p < 1 \cdot 10^{-2}$, ***- $p < \cdot 10^{-3}$.

DOI: https://doi.org/10.7554/eLife.40675.019

Across different genes in the mouse genome, *Kcns1* contained clock CpG sites present in both WLMT and blood DNAm clocks; this gene was represented by multiple CpG sites: 6 in the blood clock and 11 in the WLMT. It has been shown recently by pyrosequencing that the CpG sites of this genomic region have a high correlation with age (*Han et al., 2018*). Interestingly, CpG sites with higher weights on the methylation clock based on RRBS also showed higher correlation with age measured with pyrosequencing. For example, the CpG site located at the position chr2:164168131 had the highest weight in the WLMT clock in this region and was also among the top 5% of the most important CpG sites representing this clock; methylation level on this CpG site was also strongly correlated with age with $R^2 = 0.81$.

To study robustness of the WLMT clock, we have also developed 100 new individual clocks based on the same set of samples used for WLMT construction. To construct every individual clock among 100, this set was split differently into 80% training and 20% test subsets with stratification based on tissues. Thus, the procedure was the same, but samples in the training and test sets varied. Next, we calculated how many times each of the 435 WLMT clock sites appeared in the newly constructed 100 clocks (*Figure 2-figure supplement 3A*). Remarkably, the five sites which are found in both WLMT and blood clocks were significantly more represented than the other sites ($p = 3.6 \cdot 10^{-4}$, Wilcoxon test). We have then tested whether the sites with larger absolute weight and thus higher impact in WLMT appeared in these 100 clocks more often (*Figure 2-figure supplement 3B*). Even though the correlation was not strong with $R^2 = 0.37$, it was highly significant ($p = 3.3 \cdot 10^{-44}$). Interestingly, a CpG site with the highest absolute weight, chr8:110168311, was found in all new clocks as well as in the blood clock. The site is located in the promoter region of CALB2. Three other sites from the same region were found in the WLMT clock and had high weights, making this region a good candidate for a targeted approach (*Han et al., 2018*). The CpG site chr2:164168131, which appeared in WLMT and the 3-site pyrosequencing-based DNAm clock (*Han et al., 2018*) was found in 95 clocks out of 100. At the same time, the vast majority of WLMT sites were detected in less than a half of the new 100 clocks, supporting the notion of global methylation change during aging that allows to select different subsets of CpG sites for DNAm clocks.

Overall, we found that single-tissue-based clocks generally reflected tissue-specific age-dependent changes in DNA methylation and were simultaneously more precise when applied to the corresponding tissue samples than multi-tissue clocks. At the same time, while multi-tissue clocks were biased toward tissues better represented in the training sets, they nonetheless better reflected systemic aging. In this regard, both tissue-specific and multi-tissue DNAm clocks should find application in future studies.

## Discussion

Most of the CpG sites with positive weight in WLMT had low DNAm levels in early life and increased with age (*Figure 2—figure supplement 4*), and the opposite was true for the clock sites with negative weight (*Figure 2—figure supplement 5*). A similar pattern of DNAm drifting towards intermediate methylation states was shown for DNAm remodeling during aging (*Sziráki et al., 2018*). Taken together, these observations support the idea of the DNAm clock being a representation of some global, systemic age-dependent changes (*Horvath and Raj, 2018*) in DNA methylation rather than being a tool to identify a small set of key genomic positions, methylation status of which determines the methylation age.

DNA methylation is a relatively new evolutionary mechanism to control gene expression. Chordates lack CpG islands (CGI), fishes and amphibians have only a small fraction of transcription start sites (TSS) containing CGI, and most of the TSS are associated with CGIs in warm-blooded vertebrates (*Sharif et al., 2010*). DNA methylation patterns also show strong tissue specificity (*Lokk et al., 2014*). As a consequence, DNA methylation profiles change with age differently in individual tissues (*Maegawa et al., 2010*). Thus, construction of a multi-tissue DNA methylation clock cannot be reduced to a simple superposition of tissue-specific clocks, signifying the presence of a general pattern of aging in the DNA methylome. Furthermore, most CpG sites belonging to tissue-specific DNAm clocks do not contribute to the constructed multi-tissue clock.

Even though DNAm clocks presently seem to be the best available estimators of age, the downside of these tools is that they are not universal and must be recreated or at least adjusted for individual species. Interestingly, this technique was first developed for human samples, whereas its application

to model organisms lagged. When the first multi-tissue DNAm clock was developed (*Horvath, 2013*), thousands of samples processed with the same technology were already available, and the original human multi-tissue clock was built based on 8000 samples representing 82 datasets (*Horvath, 2013*).

The situation in mice has been and remains drastically different. First, the number of samples sequenced and available in the public domain is lower by more than an order of magnitude. Second, these samples originate from a limited number of studies (*Petkovich et al., 2017*), and the associated methylomes are thus subject to strong batch effects, which are hard to separate from the tissue-specific effects. Having additional sources of data to construct DNAm clocks would have helped to deal with the biases introduced by individual laboratories and studies. Third, even though all mouse samples have been bisulfite sequenced, the actual experimental protocols used in different studies varied significantly. Performance of mouse clocks is expected to improve with the increase in the number of bisulfite sequenced mouse samples and the addition of new publicly available datasets. However, even the already available clocks can be used to examine the effects of candidate lifespan-extending interventions and address diverse questions in aging research.

Methylation clocks are based on chronological ages of samples. Which of them can help estimate biological age requires further analyses. Testing if the clocks predict survival differences is also a desirable future project, but it could not have been done with the currently existing data. On the other hand, some interventions seem to affect biological age. The ability of the clocks to reflect these interventions would support the idea that methylation clocks may measure biological ages. Besides, clocks with better performance on control samples are better in detecting the effects of interventions. It can be seen for the blood clock: it has the lowest MAE among all clocks on the untreated wild type blood samples (*Figure 3E*). At the same time, it can detect significant changes in lifespan-extending interventions even in the cases where the other clocks cannot do it with the given sample size (*Figure 5A,B,E*).

The existing mouse clocks vary in the tissues they are based on as well as in the models that convert weighted DNAm levels to methylation age. WLMT and liver clocks should be used as linear models, while YOMT and blood clock used a different approach, when DNAm data is first converted into a DNAm score, which then requires an additional transformation to reflect age (*Supplementary file 3*). Another difference between the clocks is the way age was integrated into the models. Chronological age was directly used to construct the WLMT clock, while log-transformed chronological age was used in models based on blood, liver and YOMT, making these clocks more tuned towards young ages. The distributions of weights look similar for all four methylation clocks (*Figure 2-figure supplement 6*), with weight distributions being zero-symmetric ($p>0.2$ for all clocks, two-tailed Mann–Whitney U test applied to positive weights and absolute values of negative weights) and the YOMT distribution having larger variance ($p<10^{-17}$, Bartlett's test).

The most popular method to measure DNA methylation at many loci, RRBS, is an enrichment method. One of its downsides is the fact that some sites covered in one sample might not be covered at all in another. We directly faced this issue in the present work and, even though only samples with more than 2 million sites covered were selected for training the WLMT clock, these 2 million sites represented mere 10% of all possible CpG sites covered in at least one sample. This brings a question about how to best apply RRBS-based clocks to future samples. The median number of WLMT clock sites covered in the samples not used for the construction of the clock (and, thus, not pre-selected) was 435, equal to the total number of the sites the WLMT clock is based on. The testing set included the independent dataset (*Reizel et al., 2018*) and intervention tests but not the dataset with a low number of covered sites (*Schillebeeckx et al., 2013*). None of the clocks comparatively studied here had good clock site coverage in this low-coverage dataset, but WLMT was leading with 75% of the clock sites utilized. Less than 90% of WLMT sites were covered in all WGBS datasets. The same is true for YOMT and blood clock – the other two clocks purely based on RRBS technology. Interestingly, blood clock sites were less represented in the datasets which were not used for training the blood clock. The blood clock is the only one among studied here for which the median site coverage in the independent dataset (*Reizel et al., 2018*) was below 100% (87 out of 90 sites). This can be explained by the fact that the blood clock was trained on the dataset created under the same conditions, while the other clocks were trained on the overlap of multiple datasets, thus neglecting CpGs in the positions which have experiment-to-experiment variability.

Comparing the clocks, we noticed an interesting phenomenon – both multi-tissue clocks and the blood clock reflect changes of methylation age in relation to changes in chronological age in all

tested tissues, even though those changes are detected better in some tissues than in the others. Performance of the blood clock decreases in following order: liver, lung, heart and cortex. But this is not the case for the liver clock, another single tissue-based clock. While it performs well on the liver samples, it fails to reflect changes in any other tissue (*Figure 3—figure supplement 2*). A possible explanation for this difference can be based on the fact that blood has a better representation of multiple cell types than liver mostly consisting of one cell type – hepatocytes which are polyploid (and as such the majority of liver DNA is the hepatocyte DNA (*Duncan et al., 2010*)). It is also possible that blood better represents organismal aging than liver.

Overall, we created an easily applicable tool to estimate the DNAm age of mouse samples of different tissues, which works across the entire lifespan of these animals. The constructed multi-tissue DNAm clock can also be applied to mice of both sexes and may be a preferred method to study aging of tissues lacking tissue-specific clocks.

## Materials and methods

### Datasets

We used tissues from the animals analyzed in our previous study (*Petkovich et al., 2017*) (C57BL/6N mice; RRID: MGI:5913915). To minimize the effect of brain heterogeneity (*Lister et al., 2013*), we homogenized the right halves of cortex prior to sampling. For liver, heart and lung sample, we cut 3–10 mg pieces while preserving frozen tissues. DNA was extracted using DNeasy Blood and Tissue Kit from Qiagen and eluted in 200 µl of 10 mM Tris-HCl buffer, pH 8.0. Then, 2 µl of RNase A (Life Technologies) was added to each sample. Samples were incubated at room temperature for 5 min, and DNA was prepared by using Genomic DNA Clean and Concentrator-10 (Zymo). DNA was eluted in 50 µl of TE buffer (10 mM Tris-HCl, 0.1 mM EDTA, pH 8.0) and quantified using a Qubit 2.0 (Life Technologies). RRBS sample preparation was performed using 100 ng of DNA following the protocol from *Petkovich et al. (2017)*. Each library was made of 3–6 samples. In order to avoid age- or tissue-related batch effect, we grouped samples into the libraries distributing each group of samples into different libraries (*Supplementary file 5*). The libraries were sequenced with Illumina HiSeq2500, PE150 with average 44M (*Figure 1-figure supplement 4*) reads per sample. 20% of mouse genomic DNA library was spiked in to compensate for low complexity of the libraries. The data obtained in this study were deposited to GEO with the accession number GSE121141.

### Data processing

In addition to data generated by our laboratory, we downloaded publicly available datasets of mouse DNAm profiles, including GSE93957, GSE60012, GSE52266, GSE80761, GSE45361, SRP069120, GSE89273, GSE89274, GSE84573, GSE70538, GSE80672. Since data in these projects were processed differently, which could in principle affect estimated methylation levels, we utilized raw reads and applied the same pipeline to all samples. The reads were trimmed using Trim Galore v0.4.1 (RRID: SCR_011847) with the parameter –rrbs for all samples, –paired for the paired-end sequenced. For the samples from GSE93957, we have also added – three_prime_clip_R1 15 –three_prime_clip_R2 15 options to remove unique molecular indices. The filtered reads were mapped to the mouse genome (GRCm38/mm10) using Bismark v0.15.0 (RRID: SCR_005604) with the default parameters to obtain methylation levels and coverage per site. Sites in the mitochondrial genome and sex chromosomes were removed. The same pipeline was applied to the independent dataset GSE85251.

### DNAm clock construction

We have created a dataset of 416 samples which met the following conditions: (1) all included mice were wild-type animals, (2) untreated and not used as an intervention control, (3) number of CpG sites covered by sequencing in every sample was greater than $2 \times 10^6$, and (4) number of the overlapping sites in all samples was greater than 100,000. We applied a soft coverage cutoff, excluding CpG sites which had <5x coverage in more than 90% of samples. This resulted in obtaining 193,439 CpG sites included in all DNA methylomes. Custom scripts for data filtering and processing were written in R and python programming languages. They were deposited to GitHub under GNU General Public License: https://github.com/gr-meer/WLMT (*Meer, 2018*; copy archived at https://github.com/elifesciences-publications/WLMT). The relation of CpG sites to CpG islands and genes

was annotated using annotatr package from Bioconductor (*Cavalcante and Sartor, 2017*) (RRID: SCR_006442). ElasticNet regression was performed using the python scikit-learn package (*Pedregosa, 2012*) (RRID: SCR_002577). Stratification based on tissue types was applied to splitting samples into training (80%) and test (20%) set and during 10-fold cross validation. Venn diagram was created using Venny 2.1 (http://bioinfogp.cnb.csic.es/tools/venny/).

## Additional information

### Funding

| Funder | Grant reference number | Author |
|---|---|---|
| National Institute on Aging | AG021518 | Vadim N Gladyshev |
| National Institute on Aging | AG047200 | Vadim N Gladyshev |

The funders had no role in study design, data collection and interpretation, or the decision to submit the work for publication.

### Author contributions

Margarita V Meer, Conceptualization, Resources, Data curation, Formal analysis, Validation, Investigation, Writing—original draft, Writing—review and editing; Dmitriy I Podolskiy, Conceptualization, Methodology, Writing—review and editing; Alexander Tyshkovskiy, Formal analysis, Investigation, Methodology, Writing—review and editing; Vadim N Gladyshev, Conceptualization, Supervision, Funding acquisition, Writing—original draft, Project administration, Writing—review and editing

### Author ORCIDs

Margarita V Meer (iD) http://orcid.org/0000-0001-8249-7097
Vadim N Gladyshev (iD) http://orcid.org/0000-0002-0372-7016

### Decision letter and Author response

Decision letter https://doi.org/10.7554/eLife.40675.046
Author response https://doi.org/10.7554/eLife.40675.047

## Additional files

### Supplementary files

• Supplementary file 1. Sample information.
DOI: https://doi.org/10.7554/eLife.40675.020

• Supplementary file 2. Information on samples sequenced by RRBS in the study and included in the analysis.
DOI: https://doi.org/10.7554/eLife.40675.021

• Supplementary file 3. Mouse DNAm clocks.
DOI: https://doi.org/10.7554/eLife.40675.022

• Supplementary file 4. Genes overlapping in pairs of clocks.
DOI: https://doi.org/10.7554/eLife.40675.023

• Supplementary file 5. Distribution of samples across libraries.
DOI: https://doi.org/10.7554/eLife.40675.024

### Data availability

Sequencing data have been deposited in GEO under accession code GSE121141

The following dataset was generated:

| Author(s) | Year | Dataset title | Dataset URL | Database and Identifier |
|---|---|---|---|---|
| Meer MV, Podolskiy DI, Tyshkovskiy AE, | 2018 | DNA methylation of mouse tissues | https://www.ncbi.nlm. nih.gov/geo/query/acc. | NCBI Gene Expression Omnibus, |

| | | cgi?acc=GSE121141 | GSE121141 |
|---|---|---|---|
| Gladyshev VN | | | |

The following previously published datasets were used:

| Author(s) | Year | Dataset title | Dataset URL | Database and Identifier |
|---|---|---|---|---|
| Stubbs TM, von Meyenn F, Katrien-Stark A, Krueger F, Reik W | 2017 | Multi-tissue DNA methylation age predictor in mouse | https://www.ncbi.nlm.nih.gov/geo/query/acc.cgi?acc=GSE93957 | GEO, GSE93957 |
| Reizel Y, Spiro A, Sabag O, Skversky Y, Hecht M, Keshet I, Berman BP, Cedar H | 2014 | Gender-specific post-natal demethylation and establishment of epigenetic memory | https://www.ncbi.nlm.nih.gov/geo/query/acc.cgi?acc=GSE60012 | GEO, GSE60012 |
| Petkovich DA, Podolskiy DI, Lobanov AV, Gladyshev VN | 2016 | Using DNA methylation profiling to evaluate biological age and longevity interventions | https://www.ncbi.nlm.nih.gov/geo/query/acc.cgi?acc=GSE80672 | GEO, GSE80672 |
| Cannon MV, Buchner DA, Hester J, Miller H, Sehayek E, Nadeau JH, Serre D | 2013 | Maternal Nutrition Induces Pervasive Gene Expression Changes but no Detectable DNA Methylation Differences in the Liver of Adult Offspring [RRBS] | https://www.ncbi.nlm.nih.gov/geo/query/acc.cgi?acc=GSE52266 | GEO, GSE52266 |
| Zhang C, Edepli KS, Lujambio A | 2016 | Genome-wide DNA methylation profiling of mouse liver | https://www.ncbi.nlm.nih.gov/geo/query/acc.cgi?acc=GSE80761 | GEO, GSE80761 |
| Schillebeeckx M, Mitra RD | 2013 | Laser Capture Microdissection-Reduced Representation Bisulfite Sequencing (LCM-RRBS) maps changes in DNA methylation associated with gonadectomy-induced adrenocortical neoplasia in the mouse | https://www.ncbi.nlm.nih.gov/geo/query/acc.cgi?acc=GSE45361 | GEO, GSE45361 |
| Cole JJ, Robertson NA, Rather MI, Adams PD | 2016 | Diverse interventions that extend mouse lifespan suppress shared age-associated epigenetic changes at critical gene regulatory regions (WGBS 1) | https://www.ncbi.nlm.nih.gov/geo/query/acc.cgi?acc=GSE89273 | GEO, GSE89273 |
| Cole JJ, Robertson NA, Rather MI, Adams PD | 2016 | Diverse interventions that extend mouse lifespan suppress shared age-associated epigenetic changes at critical gene regulatory regions (WGBS 2) | https://www.ncbi.nlm.nih.gov/geo/query/acc.cgi?acc=GSE89274 | GEO, GSE89274 |
| McCormick H, Hurs, Young P, Cropley J, Booher K, Cheung H, Giannolatou E, Suter C | 2016 | Isogenic mice exhibit sexually-dimorphic DNA methylation patterns across multiple tissues | https://www.ncbi.nlm.nih.gov/geo/query/acc.cgi?acc=GSE84573 | GEO, GSE84573 |

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
