## [Decision Letter]

Thank you for submitting your article "A whole lifespan mouse multi-tissue DNA methylation clock" for consideration by *eLife*. Your article has been reviewed by three peer reviewers, and the evaluation has been overseen by a Reviewing Editor and Jessica Tyler as the Senior Editor. The following individual involved in review of your submission has agreed to reveal his identity: Peter Adams (Reviewer #2).

The reviewers have discussed the reviews with one another and the Reviewing Editor has drafted this decision to help you prepare a revised submission.

Summary:

The reviewers appreciated the value of the whole lifespan mouse multi-tissue DNA methylation clock (WLMT clock) and agree in principle that this is suitable as an *eLife* Tools and Resources article. Some concerns were raised however, and after consultation, we have identified several places where the manuscript could be improved. Most notably, all three reviewers felt that it is important to show how well the WLMT clock performs on additional data sets that have not been pre-selected to include all of the clock CpGs. The authors also must confirm that the raw data have been deposited at GEO and provide the GEO ID or a reviewer access link.

Essential Revisions:

1) While the WLMT clock performs well on samples from the test and training sets the precision appears to be much lower on other samples. This is probably due to the fact that not all of the relevant 435 clock sites are covered in these completely independent samples (while the test set was preselected to cover all 435 sites). The authors should assess how the WLMT performs on completely independent datasets. Furthermore, the authors should describe how to adjust for missing clock sites. Otherwise the method won't be useful for other researchers to apply to new datasets. Perhaps some new datasets could be generated or publicly available datasets should be used to address this important issue.

2) A major question about the novel clock is how it differs from other clocks. There are a couple of additional analyses that should be performed to address this important question.

a) The authors should compute correlations among the different epigenetic clocks and also among their age-acceleration-residuals/delta-age values (i.e. the difference between epigenetic age and chronological age).

b) Comparison of the WLMT clock with other clocks was done with different subsets of samples since almost all samples turned out to be included into one or another clock training set. However, for a fair comparison only completely independent datasets from different tissues should be used.

3) The most important results in the paper are contained in Figure 4, and these are not very easy to interpret from the graph or the accompanying text. Most interest in epigenetic clocks is driven by the hypothesis that differences between epigenetic age and chronological age (called "delta-age" or "epigenetic age acceleration residual", depending on how computed) reflect processes of biological aging. The authors present tests of this hypothesis in Figure 4 by comparing delta-age values of mice exposed to different longevity-enhancing or mortality-risk-increasing interventions. The result of this analysis is that the blood-based clock often shows differences in delta-age values, whereas results are less consistent for other clocks. This result is not transparent from the figure nor is it all that transparent from the text. It needs to be clarified. For example, it would be easier to see the result if the intervention and control group data for each clock were presented side by side so the visual comparison suggested by the figure matched the statistical analysis performed.

4) There are several places where it was felt that the authors overstate or overinterpret their results. Please correct these as suggested:

a) The authors state that Figure 3A is "is biased towards a better performance of the other clocks because the biological ages of a combination of their test and training samples are estimated, thus artificially improving apparent performance", whereas WLMT is only measured against its test set (not its training set). I think this may underestimate the inter-lab qualitative variations in RRBS datasets. One reason the human clocks work so well across datasets is, I think, the high technical consistency of the Illumina arrays between labs. WLMT is compared to RRBS data generated largely in the same lab, while the other datasets are compared to WLMT data generated in a different lab. The inter-lab comparisons my bias against the other clocks due to qualitative variations in RRBS data between labs. RRBS in different labs likely has different coverage of different CpGs, for example. Hence, I think this statement should be moderated.

b) The authors state "DNAm clocks remain the most precise markers of biological age". I think here the authors mean "chronological" age not "biological" age. First, in the Introduction the authors support this statement by reference to measures of chronological age, not biological age. Second, there is no consensus as to how to best measure biological age, so it is an overstatement to say that DNAm clocks are the best. Chronological and biological age are not the same and the terms should not be used interchangeably.

c) The statement that "DNA methylation is a relatively new evolutionary mechanism to control gene expression; among animals, it appears in vertebrates […]" (Discussion paragraph two) is not covered by Lokk et al., 2014 and it might not be correct.

d) The authors indicated that "the constructed multi tissue DNAm clock can also be applied to mice of different genetic backgrounds" (Discussion final paragraph), but this has not been systematically analyzed in this study.

e) The authors assert their multi-tissue clock is unique from previous tissue specific clocks partly on the basis that few of the CpGs included in the multi-tissue clock are included in tissue-specific clocks (Discussion paragraph two). The elastic net run on the same data against the same criterion with slightly different parameters may select different CpGs from one run to the next. The question is whether the different CpGs are more or less statistically independent of one another. The authors should test this if they wish to make claims about the uniqueness of the CpGs in their multi-tissue clock or the discussion should be appropriately modified.

---

## [Author Response]

The reviewers appreciated the value of the whole lifespan mouse multi-tissue DNA methylation clock (WLMT clock) and agree in principle that this is suitable as an eLife Tools and Resources article. Some concerns were raised however, and after consultation, we have identified several places where the manuscript could be improved. Most notably, all three reviewers felt that it is important to show how well the WLMT clock performs on additional data sets that have not been pre-selected to include all of the clock CpGs. The authors also must confirm that the raw data have been deposited at GEO and provide the GEO ID or a reviewer access link.

We would like to thank reviewers for the great comments and insights. We carried out additional analyses including the test of the WLMT clock on other publicly available datasets. We provide more details below in response to specific comments. In addition, raw methylation data and derived methylation levels on CpG sites used for training and testing the developed clock have been deposited to GEO database with the accession number GSE121141.

Essential Revisions:1) While the WLMT clock performs well on samples from the test and training sets the precision appears to be much lower on other samples. This is probably due to the fact that not all of the relevant 435 clock sites are covered in these completely independent samples (while the test set was preselected to cover all 435 sites). The authors should assess how the WLMT performs on completely independent datasets. Furthermore, the authors should describe how to adjust for missing clock sites. Otherwise the method won't be useful for other researchers to apply to new datasets. Perhaps some new datasets could be generated or publicly available datasets should be used to address this important issue.

We have tested an independent dataset, which included heart, lung and hippocampus samples from 1- and 20-week-old mice. For the WLMT, mean absolute deviance error between methylation age and chronological age was found to be ~53 days. We also tested how well sites contributing to different studied clocks are covered in this dataset. The median coverage was equal to the total number of sites for WLMT, YOMT and liver clocks, and it was 87 (97%) for the blood clock. We included this new data in Figure 4 of the manuscript. In addition, we tested the number of sites covered for different clocks in different datasets. Even though the overlap of all samples included into our dataset was ~10% of the total number of CpG sites covered in each sample, clock sites contributing to any one of four studied clocks had a high coverage (>90%) in RRBS samples with at least 2.106 sites covered. The missing sites can be compensated for by removing them from all samples examined in the analysis. While this may affect the estimated DNAm age of control samples, it ultimately helps to avoid a situation where variations in DNAm age difference are artificially created by difference in coverage of CpGs across different samples. In the case when more than 90% of clock sites are missing, which we observe for WGBS samples, we don’t recommend relying on this clock.

2) A major question about the novel clock is how it differs from other clocks. There are a couple of additional analyses that should be performed to address this important question.a) The authors should compute correlations among the different epigenetic clocks and also among their age-acceleration-residuals/delta-age values (i.e. the difference between epigenetic age and chronological age).

To investigate how WLMT differs from the other published DNAm clocks we performed an analysis of correlation between methylation ages obtained using these clocks. For this study, we only used samples not included in any of the training sets of the published clocks. Also, more than 90% of the clock sites were covered in every used sample. We found that WLMT correlates best with the blood clock when applied to the blood samples withr=0.90, whiler=0.37, when the clocks are applied to other tissues. Similarly, the correlation coefficient for WLMT and the liver clock dropped from r=0.80on liver samples to 0.13 when applied to the other tissues. At the same time, correlation between WLMT and YOMT had an intermediate value (r=0.57). In addition, we performed a correlation analysis of the delta-age values (i.e. the difference between epigenetic age and chronological age). This analysis showed a pattern which differs from the comparison of the absolute values of the DNAm age. Correlation between WLTM and blood clock was 0.42 and 0.63 on blood and non-blood samples, respectively. Meanwhile, the liver clock correlated with WLMT with r=0.81on liver samples and r=0.63on the other samples. Correlation of the delta values of WLMT and YOMT was slightly better than on the absolute values of these clocks, with r=0.66. We included this information in the revised manuscript.

b) Comparison of the WLMT clock with other clocks was done with different subsets of samples since almost all samples turned out to be included into one or another clock training set. However, for a fair comparison only completely independent datasets from different tissues should be used.

To further compare the clocks, we analyzed a completely independent dataset that included heart, lung and hippocampus samples (Figure 4 of the revised manuscript). The WLMT clock shows slightly better performance than the other clocks overall with MAE=53 days, while MAE is equal to 58 days for YOMT, 54 days for the liver clock and 64 days for the blood clock. Interestingly, YOMT and liver clocks produced tight clusters (with very low deviations of DNAm age in each cluster corresponding to a particular chronological age) barely reflecting age change. Thus, MAE might not be the best tool to compare performance of the clocks in this case. To further investigate the comparative behavior of different clocks, we performed linear regression of estimated DNAm age to the chronological age for all four of them. Linear regressions for YOMT, blood and liver clocks showed a significant shift of slope from 1. For the WLMT this change of slope was smaller and not significant. Slopes of the regressions performed on DNAm age and chronological age were: 0.14 (YOMT), 0.16 (liver), 1.24 (WLMT), 0.45 (blood). P-values: p=1.4⋅10-39(YOMT), p=2.8⋅10-29(liver), (WLMT), (blood). We included these data in the paper.

3) The most important results in the paper are contained in Figure 4, and these are not very easy to interpret from the graph or the accompanying text. Most interest in epigenetic clocks is driven by the hypothesis that differences between epigenetic age and chronological age (called "delta-age" or "epigenetic age acceleration residual", depending on how computed) reflect processes of biological aging. The authors present tests of this hypothesis in Figure 4 by comparing delta-age values of mice exposed to different longevity-enhancing or mortality-risk-increasing interventions. The result of this analysis is that the blood-based clock often shows differences in delta-age values, whereas results are less consistent for other clocks. This result is not transparent from the figure nor is it all that transparent from the text. It needs to be clarified. For example, it would be easier to see the result if the intervention and control group data for each clock were presented side by side so the visual comparison suggested by the figure matched the statistical analysis performed.

We have modified the corresponding figure (Figure 5 in the revised manuscript) by presenting the results for control groups and interventions side by side. We have also expanded the corresponding text, which now reads as follows:

“We have applied the four mouse DNAm predictors to detect changes in DNAm age associated with application of longevity interventions (Figure 5A-F). […] Also, since RRBS-based DNAm clocks don’t have good coverage in WGBS samples, RRBS/WGBS – based liver clock is especially recommended for WGBS analysis performed on liver samples and it should be further tested on non-liver WGBS samples.”

4) There are several places where it was felt that the authors overstate or overinterpret their results. Please correct these as suggested:a) The authors state that Figure 3A is "is biased towards a better performance of the other clocks because the biological ages of a combination of their test and training samples are estimated, thus artificially improving apparent performance", whereas WLMT is only measured against its test set (not its training set). I think this may underestimate the inter-lab qualitative variations in RRBS datasets. One reason the human clocks work so well across datasets is, I think, the high technical consistency of the Illumina arrays between labs. WLMT is compared to RRBS data generated largely in the same lab, while the other datasets are compared to WLMT data generated in a different lab. The inter-lab comparisons my bias against the other clocks due to qualitative variations in RRBS data between labs. RRBS in different labs likely has different coverage of different CpGs, for example. Hence, I think this statement should be moderated.

We thank the reviewer for the comment. Indeed, variability of results between different labs, researchers and experiments is definitely more of an issue for RRBS-based projects than it is for the Illumina arrays. Following the reviewer’s suggestion, we moderated the sentence: “[…] on one hand is biased towards a better performance of the other clocks due to the fact that the methylation ages of a combination of their test and training samples are estimated, thus artificially improving apparent performance. But on the other hand, WGBS was based on samples representing all these datasets. Since RRBS is affected by where and by whom it was performed, this gives an advantage to WLMT because other clocks were tested on the datasets not represented in their original training sets.”

b) The authors state "DNAm clocks remain the most precise markers of biological age". I think here the authors mean "chronological" age not "biological" age. First, in the Introduction the authors support this statement by reference to measures of chronological age, not biological age. Second, there is no consensus as to how to best measure biological age, so it is an overstatement to say that DNAm clocks are the best. Chronological and biological age are not the same and the terms should not be used interchangeably.

While we believe that DNA methylation age is a particular reflection of a true biological age (taking for example into account the fact that DNA methylation age is reset to nearly 0 on iPS cells), we generally agree with this sentiment and modified the text accordingly, both the sentences mentioned by the reviewers and the other sections of the text throughout the manuscript.

c) The statement that "DNA methylation is a relatively new evolutionary mechanism to control gene expression; among animals, it appears in vertebrates […]" (Discussion paragraph two) is not covered by Lokk et al., 2014 and it might not be correct.

Thank you for catching this error in referencing. We added the missing reference and modified the test. Now it reads as follows: "DNA methylation is a relatively new evolutionary mechanism to control gene expression. Chordates lack CpG islands (CGI), fishes and amphibians have only a small fraction of transcription start sites (TSS) containing CGI, but most of the TSS are associated with CGIs in warm-blooded vertebrates.”

d) The authors indicated that "the constructed multi tissue DNAm clock can also be applied to mice of different genetic backgrounds" (Discussion final paragraph), but this has not been systematically analyzed in this study.

Thank you for pointing this out. We have added a comparison of WLMT performance on male and female samples, showing that there is no significant difference in DNAm age estimated for male and female samples of similar ages (p=0.3, two-tailed Mann–Whitney U test). We didn’t see a significant difference between C57BL6-derived strains (p=0.25), but the number of available RRBS samples of other mouse strains was not sufficient to perform a comprehensive analysis. Thus, we have removed the statement related to the genetic background.

e) The authors assert their multi-tissue clock is unique from previous tissue specific clocks partly on the basis that few of the CpGs included in the multi-tissue clock are included in tissue-specific clocks (Discussion paragraph two). The elastic net run on the same data against the same criterion with slightly different parameters may select different CpGs from one run to the next. The question is whether the different CpGs are more or less statistically independent of one another. The authors should test this if they wish to make claims about the uniqueness of the CpGs in their multi-tissue clock or the discussion should be appropriately modified.

Thank you for the valuable suggestions. In order to test the uniqueness of the clock CpG sites we have additionally constructed 100 clocks using elastic net regression on the same dataset varying the samples included in the training and test sets. The CpGs which overlap between the WLMT and blood clock appeared in significantly larger number of the clocks (p-value =3.6x10^-4^, Wilcoxon test). At the same time, the vast majority of the sites was observed in less than a half of these new clocks further supporting the concept of the global methylation change during aging, allowing us to select different subsets of CpG sites for DNAm clocks. We included a corresponding figure supplement (Figure 2—figure supplement 3).